# Asynchronous SGD without Gradient Delay for Efficient Distributed Training

## Abstract

Asynchronous distributed gradient descent algorithms for training of deep neural networks are usually considered as inefficient, mainly because of the Gradient delay problem. In this paper, we propose a novel asynchronous distributed algorithm that tackles this limitation by well-thought-out averaging of model updates, computed by workers. The algorithm allows computing gradients along the process of gradient merge, thus, reducing or even completely eliminating worker idle time due to communication overhead, which is a pitfall of existing asynchronous methods. We provide theoretical analysis of the proposed asynchronous algorithm, and show its regret bounds. According to our analysis, the crucial parameter for keeping high convergence rate is the maximal discrepancy between local parameter vectors of any pair of workers. As long as it is kept relatively small, the convergence rate of the algorithm is shown to be the same as the one of a sequential online learning. Furthermore, in our algorithm, this discrepancy is bounded by an expression that involves the staleness parameter of the algorithm, and is independent on the number of workers. This is the main differentiator between our approach and other solutions, such as Elastic Asynchronous SGD or Downpour SGD, in which that maximal discrepancy is bounded by an expression that depends on the number of workers, due to gradient delay problem. To demonstrate effectiveness of our approach, we conduct a series of experiments on image classification task on a cluster with 4 machines, equipped with a commodity communication switch and with a single GPU card per machine. Our experiments show a linear scaling on 4-machine cluster without sacrificing the test accuracy, while eliminating almost completely worker idle time. Since our method allows using commodity communication switch, it paves a way for large scale distributed training performed on commodity clusters.

## 1 Introduction

Distributed training of deep learning models is devised to reduce training time of the models. Synchronous distributed SGD methods, such as Chen et al. (2016) and You et al. (2017), perform training using mini-batch size of several dozens of thousands of images. However, they either require expensive communication switch for fast gradient sharing between workers or, otherwise, introduce a high communication overhead during gradient merge, where workers are idle waiting for communicating gradients over communication switch.

Distributed asynchronous SGD methods reduce the communication overhead on one hand, but usually introduce *gradient delay problem* on the other hand, as described in Chen et al. (2016). Indeed, usually in an asynchronous distributed approach, a worker $w$ obtains a copy of the central model, computes a gradient on this model and merges this gradient back into the central model. Note, however, that since the worker obtained the copy of the central model till it merges its gradient back into the central model, other workers could have merged their gradients into the central model. Thus, when the worker $w$ merges its gradient into a central model, that model may have been updated and, thus, the gradient of the worker $w$ is *delayed*, leading to gradient delay problem. We will refer to algorithms that suffer from gradient delay problem as *gradient delay algorithms*, e.g. Downpour SGD Dean et al. (2012).

As our analysis reveals in Section 3, the quantity that controls the convergence rate of an asynchronous distributed algorithm, is *maximal pairwise distance* – the maximal distance between local models of any pair of workers at any iteration. Usually gradient delay algorithms do not limit this distance and it may depend on the number of asynchronous workers, which may be large in large clusters. This may explain their poor scalability, convergence rate and struggle to reach as high test accuracy as in synchronous SGD algorithms, as experimentally shown in Chen et al. (2016).

While Elastic Averaging SGD Zhang et al. (2015) is also a gradient delay algorithm, it introduces a penalty for workers, whose models diverge too far from the central model. This, in turn, helps to reduce the maximal pairwise distance between local models of workers and, thus, leads to better scalability and convergence rate. In contrast, our analysis introduces staleness parameter that directly controls the maximal pair distance of the asynchronous workers.

Our analysis builds on the work of Zinkevich et al. (2009b), who studied convergence rate of gradient delay algorithms, when the maximum delay is bounded. They provided analysis for Lipschitz continuous losses, strongly convex and smooth losses. While they show that bounding staleness can improve convergence rate of a gradient delay algorithm, in their algorithm each worker computes exactly one gradient and is idle, waiting to merge the gradient with PS model and download the updated PS model back to the worker.

The main contributions of this paper are the following. We present and analyze a new asynchronous distributed SGD method that both reduces idle time and eliminates gradient delay problem. Our main theoretic result shows that an asynchronous distributed SGD algorithm can achieve convergence rate as good as in sequential online learning. We support this theoretic result by conducting experiments that show that on a cluster with up to 4 machines, with a single GPU per machine and a commodity communication switch, our asynchronous method achieves linear scalability without degradation of the test accuracy.

## 2    ALGORITHM DESCRIPTION

Below we describe two algorithms. We will use the terms *model* and *parameter vector* to refer to the collection of trainable parameters of the model. Each worker in these algorithms starts computing gradients from a copy of a PS model and, prior to merging with the central model at PS, the worker can compute either a single gradient or several gradients, advancing the local model using all these gradients. In the sequel we will use term *model update* of a worker – the difference between the last local model prior to the merge and the latest copy of PS model, from which the worker started computing gradients.

### 2.1    ASYNCHRONOUS SGD, WITHOUT GRADIENT DELAY

The key idea of our approach is to reduce gradient delay, as described in Algorithm 1. To achieve this goal, the merge process waits till *each* worker has at least one gradient computed locally. Then, model updates from all the workers are collected and averaged and the average model update is used to update the model at PS. This is the synchronous part of our hybrid algorithm. In this way we replace gradient delay by averaging gradients, which is widely used as a technique to increase mini-batch size.

Workers are allowed to compute gradients asynchronously to each other and to the merge process to reduce wait times, when workers are idle. This is the asynchronous part of our hybrid algorithm.

Furthermore, the idea of computing several gradients to form a model update is used to *hide* communication overhead of gradient merge with useful computation of gradients at the workers. Master starts with computing the initial version of the model in line 27 and provides it for transferring to workers. Then Master waits till each worker has computed at least one gradient and used it to advance the worker's local model. When this happens, Master instructs on transferring a model update $w.\Delta \mathbf{x}$ from each worker to PS, where model updates from all workers are averaged and the average is used to advance the PS model. Finally the updated PS model is provided to all workers.

A worker starts with assigning value 0 to each variable, except variable *staleness $s$*, which is assigned the maximal staleness $\tau$. In line 5 the worker checks if the staleness $s$ has already achieved its maximal value and in this case it waits in line 6 till an initial version of the PS model is transferred to

the worker. When the PS model is transferred to the worker, in lines 9 and 10, the worker initializes the two model variables $\mathbf{x}$ and $\mathbf{x}_{init}$ with the PS model, since at this stage model update $w.\Delta\mathbf{x}$ is still 0. Next, the worker sets staleness to 0. Lines 13-15 comprise a usual update of the local model. In line 16, the worker notifies Master on availability of a non-zero model update $w.\Delta\mathbf{x}$. Now in line 17, the worker advances the local iteration counter $i+ = 1$. It also advances staleness $s$, since with advancing the local model with one gradient, the local model gets far away from the latest version of PS model, stored in line 9 in $\mathbf{x}_{init}$, by one more gradient. In this way, the worker can perform several iterations, computing a gradient and advancing the local model in lines 13-15, as long as the current staleness $s$ does not hit the maximal staleness $\tau$.

Assume now that at some point in time, Master requests to transfer model update $w.\Delta\mathbf{x}$ to PS in line 30. Each worker receives this request in line 21 of Thread 2. In this case, the worker releases the model update in line 22 to transfer to PS to merge with the PS model and sets the local model $\mathbf{x}$ to $\mathbf{x}_{init}$, to indicate that the model update is re-initialized to 0. The value $i_{init}$ is re-initialized to $i$ to indicate that the number of gradients, accumulated in model update $\mathbf{x} - \mathbf{x}_{init}$ is 0.

---

**Algorithm 1:** Asynchronous SGD without Gradient Delay.

---

1  Worker Procedure: THREAD 1
2  Input: $\tau$-maximal staleness, $\alpha_i$-learning schedule.
3  initialization: $i = i_{init} = 0, s = \tau, \mathbf{x} = \mathbf{x}_{init} = w.\Delta\mathbf{x} = 0$;
4  **while** *true* **do**
5      **if** $s = \tau$ **then**
6          |   wait to receive model from $PS$: $PS.\mathbf{x}$;
7      **end**
8      **if** $PS.\mathbf{x}$ *is available* **then**
9          $\mathbf{x}_{init} = PS.\mathbf{x}$;
10         $\mathbf{x} = \mathbf{x}_{init} + w.\Delta\mathbf{x}$;
11         $s = i - i_{init}$;
12     **end**
13     compute gradient $g(\mathbf{x})$;
14     $\mathbf{x} = \mathbf{x} - \alpha_i g(\mathbf{x})$;
15     $w.\Delta\mathbf{x} = \mathbf{x} - \mathbf{x}_{init}$
16     notify Master on $w.\Delta\mathbf{x}$;
17     $i = i + 1, s = s + 1$;
18 **end**
19 Worker procedure: THREAD 2
20 **while** *true* **do**
21     wait for a Master request to transfer $w.\Delta\mathbf{x}$
22     provide $w.\Delta\mathbf{x}$ to transfer to PS.
23     $\mathbf{x}_{init} = \mathbf{x}, i_{init} = i$
24 **end**
25 Master Procedure.
26 initialization: $W$ – the set of workers, $PS.\mathbf{x}$ = initial model;
27 provide $PS.\mathbf{x}$ to workers;
28 **while** *not stop condition* **do**
29     wait for each worker $w$ to provide $w.\Delta\mathbf{x}$.
30     request each worker $w$ to transfer its model update $\Delta x$.
31     **for** *each worker $w$* **do**
32         $PS.\mathbf{x} = PS.\mathbf{x} + \frac{1}{|W|} w.\Delta\mathbf{x}$
33     **end**
34     provide $PS.\mathbf{x}$ to workers;
35 **end**

---

When the worker discovers in line 8 of THREAD 1, that there is a new version of PS model available locally, it stores this model in $\mathbf{x}_{init}$, advances the local model $\mathbf{x}$ from $\mathbf{x}_{init}$ using the model update $w.\Delta\mathbf{x}$ and sets staleness to the number of gradients $i - i_{init}$, used to compute the local update.

Setting staleness to this value indicates that the local model diverges from the latest, locally available PS model, by this number of locally computed gradients.

## 2.2 SIMPLIFIED ASYNCHRONOUS SGD WITHOUT GRADIENT DELAY

Algorithm 1 is rather hard to analyze. To simplify the analysis, we formulate Algorithm 2 that is a simplified version of Algorithm 1. Algorithm 2 incorporates the most of asynchrony of Algorithm 1: each worker computes locally multi-gradient model update, workers compute their model updates in parallel to each other and to synchronous merge of previous model updates from all the workers. This means that if either the communication between workers is fast enough or staleness is large enough, worker idle time can be eliminated completely. The only difference between the algorithms is that Algorithm 2 synchronizes the time, when workers receive a new model from PS and provide their model updates to PS to merge with the PS model.

More specifically, Algorithm 2 works in cycles of $\frac{\tau}{2}$ iterations, where $\tau$ is the maximal staleness. To simplify notation, we denote $\beta$ the cycle length, i.e. $\beta = \frac{\tau}{2}$. At the end of each cycle, e.g. at iteration $j\beta$, all workers synchronize with PS: each worker acquires the copy of a new PS model and provides its accumulated model update to PS. If PS does not have enough time to merge model updates from the previous cycle and transfer the new model to the workers, workers wait for the new PS model in line 6. When a worker receives the new PS model, it computes the model update $w.\Delta\mathbf{x}$ in line 7 and provides it to PS in line 8. Note that due to the cycle length of $\beta$ iterations, model update $w.\Delta\mathbf{x}$ is computed using $\beta$ locally computed gradients. Now, in line 9, the worker advances the new PS model using the model update and sets the resulting model to $\mathbf{x}_{init}$ and $\mathbf{x}$. Note that at this point in time, at each worker, the parameter vector $\mathbf{x}$ is sum of the new PS model, which is the same in all the workers, and the model update $w.\Delta\mathbf{x}$ that is computed using $\beta$ gradients, computed locally in the worker. This means that the worker can perform $\beta$ additional iterations before hitting the staleness boundary of $\tau$.

Now, after the synchronization with PS, each worker computes $\beta$ gradients locally and uses them to advance the local model in lines 11-13. At this point in time, each worker hits the staleness boundary of $\tau$. In parallel with this computation of model update in each worker, in lines 20-24, Master transfers model updates that workers provided to PS in iteration $j\beta$, merges them with PS model and provides the new model to the workers. When each worker finishes iteration $(j + 1)\beta$, the current cycle completes and, again, synchronization starts between workers and PS.

Note that workers in Algorithm 2 may be idle, waiting for the new PS model in line 6, only if the communication between workers and PS is slow and does not allow transferring model updates from workers to PS, merge them with the PS model and transfer the new PS model back to the workers before workers complete computing $\beta$ gradients. Also note that further increasing staleness, reduces or completely eliminates worker idle time.

## 3 ANALYSIS

In this section, we analyze Algorithm 2. In the analysis we adopt the notation of Zinkevich et al. (2009b). In the proof of Lemma 3.4 and of Theorem 3.5, we use derivation similar to Zinkevich et al. (2009b), while adding terms, which are relevant for our specific algorithms. The most of the analysis is our own contribution. We will point out to parts that we borrow from prior research work.

Our main result is Theorem 3.9, where we show that the convergence rate of Algorithm 2 is $O(\tau^2 + \sqrt{T})$. The algorithm has two phases. In the first phase, during the early iterations, the asynchronous training leads to a slow-down, expressed in the term $\tau^2$. However, in the second phase, during later iterations, the asynchronous training appears to be harmless. We show that, if $T \gg \tau$, the convergence rate of Algorithm 2 is as good as in sequential online learning.

Note that the convergence rate, stated in Theorem 3.9, assumes that the standard deviation of gradient computation is inversely proportional to the maximal staleness $\tau$. In practice, this means that for each given value of maximal staleness $\tau$, one needs to set the size of mini-batch at each worker so that the standard deviation of gradient computation gets below $\frac{1}{\tau}$.

---

**Algorithm 2:** Simplified Asynchronous SGD without Gradient Delay.

---

**1** Worker Procedure.
**2** Input: $\tau$-even maximal staleness, $\beta = \frac{\tau}{2}$ and $\alpha_i$-learning schedule.
**3** initialization: $i = i_{init} = 0$, $\mathbf{x} = \mathbf{x}_{init} = w.\Delta\mathbf{x} = 0$;
**4** **while** *true* **do**
**5**  **if** $i \equiv 0 \pmod{\beta}$ **then**
**6**    wait to receive model from $PS$: $PS.\mathbf{x}$ ;
**7**    $w.\Delta\mathbf{x} = \mathbf{x} - \mathbf{x}_{init}$;
**8**    provide $w.\Delta\mathbf{x}$ to transfer to $PS$;
**9**    $\mathbf{x} = \mathbf{x}_{init} = PS.\mathbf{x} + w.\Delta\mathbf{x}$;
**10**  **end**
**11**  compute gradient $g(\mathbf{x})$;
**12**  $\mathbf{x} = \mathbf{x} - \alpha_i g(\mathbf{x})$;
**13**  $i = i + 1$;
**14** **end**
**15** Master Procedure.
**16** initialization: $W$ – the set of workers, $PS.\mathbf{x}$ = initial model;
**17** provide $PS.\mathbf{x}$ to workers;
**18** **while** *not stop condition* **do**
**19**  wait for each worker $w$ to provide $w.\Delta\mathbf{x}$.
**20**  request each worker $w$ to transfer its model update $\Delta x$.
**21**  **for** *each worker $w$* **do**
**22**    $PS.\mathbf{x} = PS.\mathbf{x} + \frac{1}{|W|} w.\Delta\mathbf{x}$
**23**  **end**
**24**  provide $PS.\mathbf{x}$ to workers;
**25** **end**

---

Now we compare our main result, stated in Theorem 3.9 with the main result of Zinkevich et al. (2009a), stated in Theorem 8, where the convergence bound is $O(\tau^2 \log T + \sqrt{T})$. While the behavior in the second phase is the same $O(\sqrt{T})$, in the first training phase, our result $O(\tau^2)$ is better than in Zinkevich et al. (2009a) $O(\tau^2 \log T)$, since our result does not have the dependency on $T$.

Denote convex (cost) functions by $f_i : X \to \mathbb{R}$, and their parameter vector by $\mathbf{x}$. Our goal is to find a sequence of $\mathbf{x}_i$ such that the cumulative loss $\sum_i f_i(\mathbf{x}_i)$ is minimized. With some abuse of notation, we identify the average empirical and expected loss both by $f^*$. This is possible, simply by redefining $p(f)$ to be the uniform distribution over $F$. Denote by

$$f^* = \frac{1}{F} \sum_i f_i(\mathbf{x}) \ \ or \ f^*(\mathbf{x}) = E_{f \sim p(f)}[f(\mathbf{x})] \ \ and \ \ x^* = \arg\min_{\mathbf{x} \in X} f^*(\mathbf{x}) \ .$$

the average risk. We assume that $x^*$ exists (convexity does not guarantee a bounded minimizer) and that it satisfies $||\mathbf{x}^*|| \leq R$ (this is always achievable, simply by intersecting $X$ with the unit-ball of radius $R$).

We remind that in Algorithm 2, in each cycle each worker computes an update to its local parameter vector, based on $\beta$ locally computed gradients and PS averages these per-worker updates from all the workers to update its own parameter vector.

In Algorithm 2 at time $t$, each worker $w$ computes the gradient of the same function $f$ on its own parameter vector $\mathbf{x}_{t,w}$ and its own mini-batch. We denote this function at worker $w$ and time $t$ $f_{t,w}$ and denote the gradient of this function, computed at the local parameter vector $\mathbf{x}_{t,w}$, $g_{t,w} = \nabla f_{t,w}(\mathbf{x}_{t,w})$.

For the analysis, we define the global parameter vector at time $t$ (as opposite to per-worker parameter vector $\mathbf{x}_{t,w}$) as average of per-worker parameter vectors

$$\mathbf{x}_t = \frac{1}{|W|} \sum_{w \in W} \mathbf{x}_{t,w} \ . \tag{1}$$

Also we denote

$$f_t(\mathbf{x}_t) = \frac{1}{|W|} \sum_{w \in W} f_{t,w}(\mathbf{x}_t) \, . \tag{2}$$

We assume that each $f_{t,w}$, and thus $f_t$, is convex, and subdifferentials of $f_t$ are bounded $||\nabla f_t(\mathbf{x})|| \leq L$ by some $L > 0$. Denote by $\mathbf{x}^*$ the minimizer of $f^*(\mathbf{x})$. We want to find a bound on the regret $R$, associated with a sequence $X = \mathbf{x}_1, ..., \mathbf{x}_T$ of parameter vectors

$$R[X] = \sum_{t=1}^{T} f_t(\mathbf{x}_t) - f_t(\mathbf{x}^*) \, . \tag{3}$$

Such bounds can be converted into bounds on the expected loss, as in Shalev-Shwartz & Singer (2007), for an example. Note that with the definitions (1) and (2), the regret in (3) is well-defined. We denote

$$\tilde{g}_t = \nabla f_t(\mathbf{x}_t) = \frac{1}{|W|} \sum_{w \in W} \nabla f_{t,w}(\mathbf{x}_t) \, . \tag{4}$$

This is how a gradient is computed in synchronous distributed SGD algorithms. Since all $f_t$ are convex, we can upper bound $R[X]$ via

$$R[X] \leq \sum_{t=1}^{T} < \nabla f_t(\mathbf{x}_t), \mathbf{x}_t - \mathbf{x}^* > = \sum_{t=1}^{T} < \tilde{g}_t, \mathbf{x}_t - \mathbf{x}^* > \, . \tag{5}$$

Let us define a distance function between $\mathbf{x}$ and $\mathbf{x}'$: $D(\mathbf{x}||\mathbf{x}') = \frac{1}{2}||\mathbf{x} - \mathbf{x}'||^2$. In Algorithm 2, each worker at the beginning of cycle $i$, i.e. at time $i\beta$, receives the copy of a new PS parameter vector. In Lemmas 3.1 – 3.3 we study properties of Algorithm 2 at this point in time. In Lemma 3.1, we show that the copy of PS model at time $i\beta$, equals to the average of local parameter vectors of workers from iteration $(i-1)\beta$, which, according to (1), equals to $\mathbf{x}_{(i-1)\beta}$.

**Lemma 3.1.** *The copy of a new PS parameter vector that each worker receives at iteration $(i+1)\beta$, equals to the average of local parameter vectors of workers from iteration $i\beta$*

$$PS.\mathbf{x}_{(i+1)\beta} = \frac{1}{|W|} \sum_{w \in W} \mathbf{x}_{w,(i+1)\beta} = \mathbf{x}_{(i+1)\beta} \, . \tag{6}$$

The proof may be found in Appendix A. Next, when a worker receives at iteration $i\beta$ the copy of a new PS parameter vector, which, according to Lemma 3.1, equals to $\mathbf{x}_{(i-1)\beta}$, it adds to this parameter vector its latest model update $\mathbf{x}_{w,i\beta} \leftarrow \mathbf{x}_{(i-1)\beta} + \Delta w.\mathbf{x}$. Note that this operation *resets* all the local parameter vectors of the worker, computed in the last cycle: the operation moves all the local parameter vectors $\{\mathbf{x}_{w,(i-1)\beta+t} | t = 0, \ldots, \beta\}$ along the vector $\mathbf{x}_{w,(i-1)\beta} - \mathbf{x}_{(i-1)\beta}$ towards the average parameter vector $\mathbf{x}_{(i-1)\beta}$:

$$\mathbf{x}_{w,(i-1)\beta+t} \leftarrow \mathbf{x}_{w,(i-1)\beta+t} + (\mathbf{x}_{w,(i-1)\beta} - \mathbf{x}_{(i-1)\beta}), \ t = 0, \ldots, \beta \, . \tag{7}$$

Lemma 3.2 shows that computing average parameter vector (1) is invariant under rest operation (7).

**Lemma 3.2.** *Computing average parameter vector, according to (1), is invariant to the reset operation (7), i.e. for each $i \in \mathbb{N}$*

$$\mathbf{x}_{i\beta+t} = \frac{1}{|W|} \sum_{w \in W} (\mathbf{x}_{w,i\beta+t} + (\mathbf{x}_{w,i\beta} - \mathbf{x}_{i\beta})) \, , \ t = 0, \ldots, \beta \, . \tag{8}$$

The proof may be found in Appendix A. Now, Lemma 3.3 bounds the distance between parameter vectors for any pair of workers after the reset operation at iteration $i\beta$.

**Lemma 3.3.** *Suppose gradients of cost functions $f_t$ are bounded $||\nabla f_t(\mathbf{x})|| \leq L$ by some $L$. Let $w, w' \in W$ be any two workers. Then after the reset operation (7), at iteration $i\beta$,*

$$||\mathbf{x}_{i\beta,w} - \mathbf{x}_{i\beta,w'}|| \leq L\tau\eta_{(i-1)\beta} \, . \tag{9}$$

The proof may be found in Appendix A. After the reset operation at iteration $i\beta$, in the next $\beta$ iterations, each worker computes $\beta$ gradients locally to advance its local parameter vector

$$\mathbf{x}_{i\beta+t,w} = \mathbf{x}_{i\beta+t-1,w} - \eta_{i\beta+t-1}g_{i\beta+t-1,w}, \ 1 \le t \le \beta \ .$$

Using this expression, we can re-write (1)

$$\mathbf{x}_{i\beta+t} = \mathbf{x}_{i\beta+t-1} - \eta_{i\beta+t-1}\frac{1}{|W|}\sum_{w\in W} g_{i\beta+t-1,w} \ . \tag{10}$$

We abuse notation and denote an average over gradients $g_{i\tau+t,w}$ as

$$\overline{g}_{i\tau+t} = \overline{g}_{i\tau+t}(\mathbf{x}_{i\tau+t}) = \frac{1}{|W|}\sum_{w\in W} g_{i\tau+t,w} \ . \tag{11}$$

Note the difference between (4) and (11). We use expression (4) as one way to define the average gradient, with which we start to bound regret in (5). In this expression, each gradient uses the same parameter vector, defined in (1), to compute its gradient. Expression (11), is another way to compute an average gradient, where each worker uses its own local parameter vector to compute gradient. With notation (11) we rewrite (10)

$$\mathbf{x}_{i\beta+t} = \mathbf{x}_{i\beta+t-1} - \eta_{i\beta+t-1}\overline{g}_{i\tau+t-1} \ , \ t = 1\ldots,\beta \ . \tag{12}$$

Next, to prove our regret bounds, we adapt Lemma 1 from Zinkevich et al. (2009b). In Zinkevich et al. (2009b), Lemma 1 is proved for an asynchronous algorithm, in which each worker computes exactly one gradient and transfers it to PS to merge with its parameter vector. We adapt this lemma to Algorithm 2, in which each worker computes $\beta$ gradients locally and then all the workers transfer their updates of their local parameter vectors to PS, which averages them and uses this average to update its own parameter vector.

**Lemma 3.4.** *For all* $\mathbf{x}^*$*, for all* $i$ *and* $0 \le t < \beta$*, if* $X = \mathbb{R}^n$*, the following expansion holds:*

$$<\mathbf{x}_{i\beta+t} - \mathbf{x}^*, \tilde{g}_{i\beta+t}> = \frac{1}{2}\eta_{i\beta+t}||\overline{g}_{i\beta+t}||^2 + \frac{D(\mathbf{x}^*||\mathbf{x}_{i\beta+t}) - D(\mathbf{x}^*||\mathbf{x}_{i\beta+t+1})}{\eta_{i\beta+t}}$$
$$+ <\mathbf{x}_{i\beta+t} - \mathbf{x}^*, \tilde{g}_{i\beta+t} - \overline{g}_{i\beta+t}> \ . \tag{13}$$

The proof may be found in Appendix A. The decomposition (13) is very similar to standard regret decomposition bounds, such as Zinkevich et al. (2009a). We add to this decomposition a new term $<\mathbf{x}_{i\beta+t} - \mathbf{x}^*, \tilde{g}_{i\beta+t} - \overline{g}_{i\beta+t}>$ to adapt the analysis to peculiarities of Algorithm 2. This term characterizes the difference between two ways to compute an average gradient at a specific time.

In Algorithm 2, at iteration $i\beta$, for each $i$, each worker starts computing gradients after it adds its local update of its local parameter vector to the copy of a common PS parameter vector and computes $\beta$ gradients locally. As workers compute more gradients in iterations $i\beta+t, t = 1,\ldots,\beta$, their local parameter vectors get more far apart from each other. However, for sufficiently small step size, the distance between local parameter vectors in different workers remains small. The key to proving our bounds is to impose further smoothness constraints on $f_t$. The rationale is quite simple: we want to ensure that small changes in $\mathbf{x}$ do not lead to large changes in the gradient. More specifically we assume that the gradient of $f_t$ is a Lipschitz-continuous function. That is,

$$||\nabla f_t(\mathbf{x}) - \nabla f_t(\mathbf{x}')|| \le H||\mathbf{x} - \mathbf{x}'|| \ . \tag{14}$$

for some constant $H$. Following Theorem 2 from Zinkevich et al. (2009b), we use Lemma 3.4 to prove Theorem 3.5. The key difference between Theorem 3.5 and Theorem 2 from Zinkevich et al. (2009b), is that we introduce a new expression and bound it to adapt the analysis to Algorithm 2.

**Theorem 3.5.** *Suppose gradients of cost functions* $f_t$ *are bounded* $||\nabla f_t(\mathbf{x})|| \le L$ *by some* $L$ *and that* $H$ *also upper-bounds the change in the gradients, as in (14). Also suppose that* $\max_{x,x'\in X} D(x||x') < F^2$*. Given* $\eta_t = \frac{\sigma}{\sqrt{t}}$*, for some constant* $\sigma > 0$*, the regret of Algorithm 2 is bounded by*

$$R[X] \le \sigma L^2\sqrt{T} + \frac{F^2}{\sigma}\sqrt{T} + 2\tau FLH(2\tau + 2\sigma\sqrt{T - 2\tau}) \ . \tag{15}$$

*Consequently, for $\sigma^2 = \frac{F^2}{2\tau L^2}$, we obtain the bound*

$$R[X] \le 4FLH\tau^2 + F(3L + 4FH)\sqrt{\tau T} . \tag{16}$$

The proof may be found in Appendix A. According to Theorem 3.5, the convergence rate of Algorithm 2 is $O(\tau^2 + \sqrt{\tau T})$. As in Zinkevich et al. (2009a), we note that this result is expected, because an adversary can rearrange training instances so that each worker receives training instances that are very similar to training instances in other workers during $\tau$ asynchronous iterations. In this case multiple workers do not perform better than a single worker and, thus, parallel algorithm is not better than a sequential one.

We will use the results of Theorem 3.5 to prove our main result in Theorem 3.9

## 3.1 Decorrelated gradient analysis

In this section we assume that training samples at different workers are drawn independently from the same underlying distribution. Also, to improve the convergence rate, we should limit the divergence of different workers from each other, as they asynchronously advance their local parameter vectors. Namely, we introduce an assumption on variance of gradients, computed in the same point at different workers. We denote $g^* = \nabla f^*$ and assume that variation of gradients at different workers, is modeled by an additive Gaussian noise $\nabla f_{t,w}(\mathbf{x}) = g^*(\mathbf{x}) + e_{t,w}$, for $e_{t,w} \sim N(\mathbf{0}, C)$, where $C$ is a covariance matrix.

We start with the following lemma that bounds the distance between parameter vectors at any two workers $w, w' \in W$ after each one of them makes $j$ asynchronous steps, starting from parameter vectors $\mathbf{x}_{i\beta,w}$ and $\mathbf{x}_{i\beta,w'}$.

**Lemma 3.6.** *Assume that variation of the gradient of the cost function is governed by an additive Gaussian noise*

$$\nabla f_{t,w}(\mathbf{x}) = g^*(\mathbf{x}) + e_{t,w} , \tag{17}$$

*for $e_{t,w} \sim N(\mathbf{0}, C)$ for covariance matrix $C$. Then for any two worker $w, w' \in W$,*

$$||\mathbf{x}_{i\beta+j,w} - \mathbf{x}_{i\beta+j,w'}|| \le ||\mathbf{x}_{i\beta,w} - \mathbf{x}_{i\beta,w'}|| \prod_{k=0}^{j-1} (\eta_{i\beta+k}H + 1)$$

$$+ \sum_{k=0}^{j-1} \eta_{i\beta+k} ||e_{i\beta+k,w} - e_{i\beta+k,w'}|| \cdot \prod_{m=k+1}^{j-1} (\eta_{i\beta+m}H + 1) . \tag{18}$$

The proof may be found in Appendix A. Lemma 3.6 bounds the distance between local parameter vectors in two different workers, as they proceed with asynchronous iterations. Lemma 3.7 uses this bound to prove a bound on the *expected value* of this difference.

**Lemma 3.7.** *Denote $var(|e_{t,w}|) = \overline{s}$. Also, in addition to the conditions of Lemma 3.6, assume that the cost functions $f_t$ are i.i.d. Then, for*

$$t \ge t_0 = (10\sigma\tau H)^2 , \tag{19}$$

*the expectation of the difference $||\mathbf{x}_{i\tau+j,w} - \mathbf{x}_{i\tau+j,w'}||$ is bounded by*

$$E||\mathbf{x}_{i\beta+j,w} - \mathbf{x}_{i\beta+j,w'}|| \le 1.2 \cdot E||\mathbf{x}_{i\beta,w} - \mathbf{x}_{i\beta,w'}|| + 1.2 \cdot \eta_{i\beta}\tau\overline{s} . \tag{20}$$

The proof may be found in Appendix A. Using (20) for $j = \beta$, before reset operation (7),

$$E||\mathbf{x}_{(i+1)\beta,w} - \mathbf{x}_{(i+1)\beta,w'}|| \le 1.2 \cdot E||\mathbf{x}_{i\beta,w} - \mathbf{x}_{i\beta,w'}|| + 1.2 \cdot \eta_{i\beta}\tau\overline{s} . \tag{21}$$

After reset operation, updates of local parameter vectors of $w$ and $w'$, computed in iterations $t = i\beta + 1, \ldots, (i+1)\beta$, are added to the common copy of a PS model. This results in reduction of RHS of (21) by $||\mathbf{x}_{i\beta,w} - \mathbf{x}_{i\beta,w'}||$, resulting in

$$E||\mathbf{x}_{(i+1)\beta,w} - \mathbf{x}_{(i+1)\beta,w'}|| \le 0.2 \cdot E||\mathbf{x}_{i\beta,w} - \mathbf{x}_{i\beta,w'}|| + 1.2 \cdot \eta_{i\beta}\tau\overline{s} . \tag{22}$$

In (22), the bound on expected distance between local parameter vectors of workers $w$ and $w'$ at iteration $(i+1)\beta$, depends on the distance at the previous cycle, at iteration $i\beta$. Lemma 3.8 develops a similar bound without the dependence on the distance from the previous cycle.

**Lemma 3.8.** *Let $i_0$ be the smallest index, s.t. $i_0\beta \geq t_0$, for $t_0$ defined in (19). Then, for*

$$j_0 = \log \frac{\overline{s}}{L} \,, \tag{23}$$

*and any $i \geq i_0 + j_0$,*

$$E||\mathbf{x}_{(i+1)\beta,w} - \mathbf{x}_{(i+1)\beta,w'}|| \leq 2 \cdot \eta_{i\beta}\tau\overline{s} \,. \tag{24}$$

The proof may be found in Appendix A. Now, when we bounded the difference between parameter vectors in different workers, we can use this bound along with the assumption (14) to bound the difference between two ways to compute an average gradient: (4) and (11). This leads to our main result – bound on the expected regret of Algorithm 2.

**Theorem 3.9.** *In addition to the conditions of Theorem 3.5, assume that the cost functions $f_t$ are i.i.d. and that variation of gradients of the cost functions is governed by an additive Gaussian law, i.e. $\nabla f_{t,w}(\mathbf{x}) = g^*(\mathbf{x}) + e_{t,w}$, for $e_{t,w} \sim N(\mathbf{0}, C)$, s.t.*

$$\tau \cdot var(e_{t,w}) \leq 1 \,. \tag{25}$$

*Given $\eta_t = \frac{\sigma}{\sqrt{t}}$ for some constant $\sigma > 0$, the expected regret of Algorithm 2 is bounded as follows*

$$ER[X] \leq 4FLH(1 + 2\sigma^2 H)\tau^2 + \left( \sigma L^2 + \frac{F^2}{\sigma} + 4FH\sigma \right) \sqrt{T} \,. \tag{26}$$

*Consequently, for $\sigma = \frac{F}{L}$,*

$$ER[X] \leq 4FLH \left( 1 + 2 \left( \frac{F}{L} \right)^2 H \right) \tau^2 + \left( 2FL + 4\frac{F^2}{L}H \right) \sqrt{T} \,. \tag{27}$$

The proof may be found in Appendix A.

## 4 EXPERIMENTAL RESULTS

In this section we provide an initial experimental support for the effectiveness of our algorithm and show discrepancy in local parameter vectors of the asynchronous workers in two approaches: averaging models updates and gradient delay. We start with the discrepancy. For this experiment we trained ResNet50 model (Szegedy et al. (2017)) on CIFAR-10 data (Krizhevsky et al.). We show the results in Figure 1, where different gradient average plots correspond to different values of staleness.

To produce a point in a gradient average plot for a given values of staleness and the number of workers, each worker starts computing an update to its local parameter vector from a parameter vector that is common to all the workers. Workers proceed computing the gradients, updating their local parameter vectors asynchronously until each worker computes the number of gradients, as the chosen value of staleness. At this point we compute the average over the last parameter vectors in the workers. Next we compute the average distance between the last parameter vectors of the workers and the average parameter vector. This average distance is plotted in Figure 1 for the corresponding values of staleness and the number of workers.

To produce a point in the gradient delay plot, we simulate the worst case in gradient delay: workers read the central parameter vector $PS.\mathbf{x}$ in a sequence. The first worker reads the initial value of $PS.\mathbf{x}$, the second after $PS.\mathbf{x}$ is advanced with one gradient, the third after $PS.\mathbf{x}$ is advanced with two gradients, etc., until the last worker read $PS.\mathbf{x}$ after it is advanced the number of times as the number workers minus 1. Each worker computes only one gradient and uses it to advance its local parameter vector. Since the last value of $PS.\mathbf{x}$ represents the most up-to-date model in the system, the average distance is now computed between this model and all the local models of the workers. This average distance is shown in the gradient delay plot.

As we see, from Figure 1, staleness is an effective tool to keep low discrepancy of local parameter vectors of asynchronous workers – the discrepancy is roughly linear in the value of the maximal staleness parameter. In contrast, in gradient delay, this discrepancy is linear in the number of workers and, thus, is limited only by the number of workers, which may be large in very large clusters.

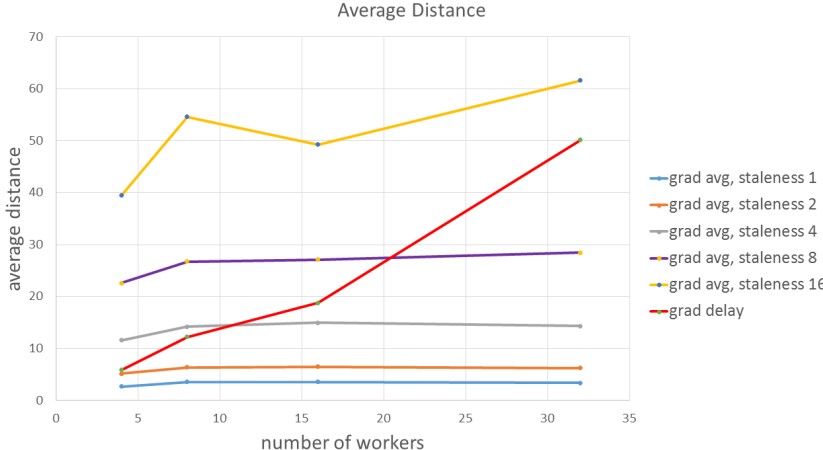

Figure 1: Average distance for various values of staleness and number of workers.

Next, we provide an initial experimental support for scalability of our algorithm. We trained GoogleNet model (Szegedy et al. (2016)) on ImageNet ILSVRC12 dataset (Russakovsky et al. (2015)) on a cluster with 4 machines. Each machine was equipped with a single Nvidia GeForce GTX TITAN X GPU card. The machines were interconnected using 1 giga bit communication switch. We used mini-batches of size 32 images and set the maximal staleness parameter to the value of 16. Also we implemented a linear scaling rule, where training with $n$ workers, we increase the learning rate $n$ times and reduce the number of iterations in each worker $n$ times. We added a linear warm-up of 50K iterations to gradually increase the base learning rate from 0.01 to $n \cdot 0.01$. After training, we test the resulting model using ImageNets own 50,000 images validation set. From Figure 2 and Figure 3, we see that our method achieves linear scalability without degradation of the test accuracy. We measured average idle time of workers and found it to be practically 0 for staleness value of at least 8.

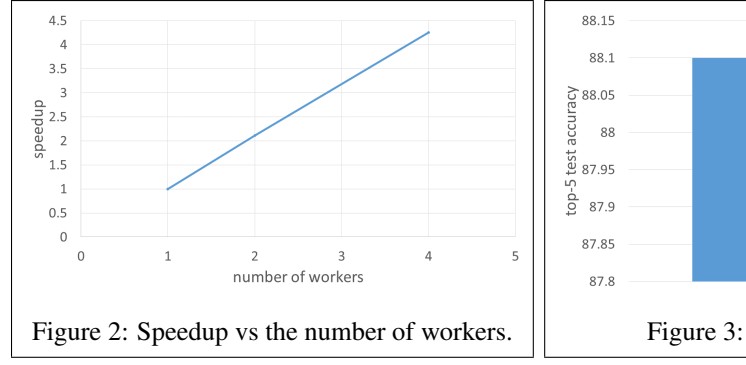

Figure 2: Speedup vs the number of workers.

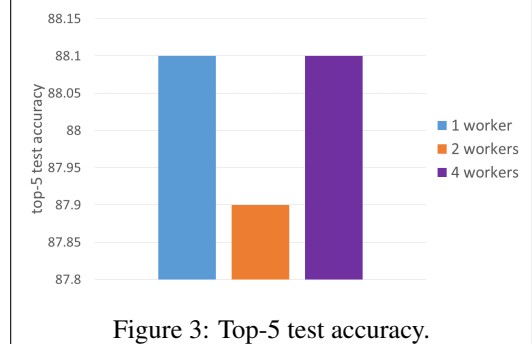

Figure 3: Top-5 test accuracy.

## 5  CONCLUSIONS

We presented a new asynchronous distributed SGD method. We show empirically that it reduces both idle time and gradient delay. We analyze the synchronous part of the algorithm, and show theoretical regret bounds.

The proposed method shows promising results on distributed training of deep neural networks. We show that our method eliminates waiting times, which allows significant improvements in run time, compared to fully synchronous setup. The very fact of efficient hiding of communication overhead opens opportunity for distributed training over commodity clusters. Furthermore, the experiments show linear scaling of training time from 1 to 4 GPU's without compromising the final test accuracy.

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

## APPENDIX A  PROOFS

Proof of Lemma 3.1.

*Proof.* We prove this statement by induction on $i$. At $i = 0$, the statement holds, since in the beginning all workers start from the common random model. Now assume that the statement holds for $i$. At iteration $i\beta$, PS collects update to local parameter vector from each worker $\mathbf{x}_{w,i\beta} - \mathbf{x}_{w,(i-1)\beta}$, averages them and uses the average to update its own parameter vector

$$PS.\mathbf{x}_{i\beta} = PS.\mathbf{x}_{(i-1)\beta} + \frac{1}{|W|} \sum_{w \in W} (\mathbf{x}_{w,i\beta} - \mathbf{x}_{w,(i-1)\beta})$$

$$= PS.\mathbf{x}_{(i-1)\beta} + \frac{1}{|W|} \sum_{w \in W} \mathbf{x}_{w,i\beta} - \frac{1}{|W|} \sum_{w \in W} \mathbf{x}_{w,(i-1)\beta} \qquad (28)$$

According to the inductive assumption, the last summand in (28) equals $PS.\mathbf{x}_{(i-1)\beta}$. □

Proof of Lemma 3.2.

*Proof.* We start with RHS of (8)

$$\frac{1}{|W|} \sum_{w \in W} \left( \mathbf{x}_{w,i\beta+t} + (\mathbf{x}_{w,i\beta} - \mathbf{x}_{i\beta}) \right) = \frac{1}{|W|} \sum_{w \in W} \mathbf{x}_{w,i\beta+t} + \frac{1}{|W|} \sum_{w \in W} \mathbf{x}_{w,i\beta} - \mathbf{x}_{i\beta} . \quad (29)$$

By the definition of average parameter vector (1), the first summand in RHS of (29) equals $\mathbf{x}_{i\beta+t}$, while the last two summands cancel each other. $\qquad \square$

Proof of Lemma 3.3.

*Proof.* At iteration $(i-1)\beta$ after reset operation (7), workers $w$ and $w'$ start updating their local parameter vectors for $\beta$ iterations, so that at the end of the cycle before the reset operation (7),

$$\mathbf{x}_{i\beta,w} - \mathbf{x}_{i\beta,w'} = \mathbf{x}_{(i-1)\beta,w} - \sum_{k=0}^{\beta-1} \eta_{(i-1)\beta+k} g_{(i-1)\beta+k,w}$$

$$- \left( \mathbf{x}_{(i-1)\beta,w'} - \sum_{k=0}^{\beta-1} \eta_{(i-1)\beta+k} g_{(i-1)\beta+k,w'} \right)$$

$$= \left( \mathbf{x}_{(i-1)\beta,w} - \mathbf{x}_{(i-1)\beta,w'} \right) - \sum_{k=0}^{\beta-1} \eta_{(i-1)\beta+k} \left( g_{(i-1)\beta+k,w} - g_{(i-1)\beta+k,w'} \right) . \quad (30)$$

The reset operation (7) at iteration $i\beta$ adds the updates of workers $w$ and $w'$, computed at iterations $(i-1)\beta+1, \ldots, i\beta$, to the common copy of PS model, so that $\mathbf{x}_{(i-1)\beta,w} - \mathbf{x}_{(i-1)\beta,w'} = 0$. Thus, after the reset operation

$$\mathbf{x}_{i\beta,w} - \mathbf{x}_{i\beta,w'} = - \sum_{k=0}^{\beta-1} \eta_{(i-1)\beta+k} \left( g_{(i-1)\beta+k,w} - g_{(i-1)\beta+k,w'} \right) . \quad (31)$$

Finally, from (31), our assumption on the size of the gradient $\nabla f_t(\mathbf{x})$ and from decreasing learning rate, during this cycle that started at iteration $(i-1)\beta$, the distance between workers $w$ and $w'$ grows at most by

$$||\mathbf{x}_{i\beta,w} - \mathbf{x}_{i\beta,w'}|| \leq L 2\beta \eta_{(i-1)\beta} = L\tau \eta_{(i-1)\beta} .$$

$\qquad \square$

Proof of Lemma 3.4.

*Proof.* We decompose our progress as follows

$$D(\mathbf{x}^*||\mathbf{x}_{i\beta+t+1}) - D(\mathbf{x}^*||\mathbf{x}_{i\beta+t})$$

$$= \frac{1}{2}||\mathbf{x}^* - \mathbf{x}_{i\beta+t} + \mathbf{x}_{i\beta+t} - \mathbf{x}_{i\beta+t+1}||^2 - \frac{1}{2}||\mathbf{x}^* - \mathbf{x}_{i\beta+t}||^2 \quad (32)$$

$$= \frac{1}{2}||\mathbf{x}^* - \mathbf{x}_{i\beta+t} + \eta_{i\beta+t}\overline{g}_{i\beta+t}||^2 - \frac{1}{2}||\mathbf{x}^* - \mathbf{x}_{i\beta+t}||^2 \quad (33)$$

$$= \frac{1}{2}\eta_{i\beta+t}^2||\overline{g}_{i\beta+t}||^2 - \eta_{i\beta+t} < \mathbf{x}_{i\beta+t} - \mathbf{x}^*, \overline{g}_{i\beta+t} >$$

$$= \frac{1}{2}\eta_{i\beta+t}^2||\overline{g}_{i\beta+t}||^2 - \eta_{i\beta+t} < \mathbf{x}_{i\beta+t} - \mathbf{x}^*, \overline{g}_{i\beta+t} - \tilde{g}_{i\beta+t} + \tilde{g}_{i\beta+t} >$$

$$= \frac{1}{2}\eta_{i\beta+t}^2||\overline{g}_{i\beta+t}||^2 - \eta_{i\beta+t} < \mathbf{x}_{i\beta+t} - \mathbf{x}^*, \tilde{g}_{i\beta+t} > + \eta_{i\beta+t} < \mathbf{x}_{i\beta+t} - \mathbf{x}^*, \tilde{g}_{i\beta+t} - \overline{g}_{i\beta+t} > .$$

To prove (33) from (32), we used (12) Dividing both sides by $\eta_{i\tau+t}$ and moving $< \mathbf{x}_{i\tau+t} - \mathbf{x}^*, \tilde{g}_{i\tau+t} >$ to the LHS completes the proof. $\qquad \square$

Proof of Theorem 3.5.

*Proof.* First we state a useful inequality: For $n$ vectors $\mathbf{a}_i$, $i = 1, \ldots, n$ (by induction on $n$):

$$\left| \sum_{i=1}^{n} \mathbf{a}_i \right| \leq \sum_{i=1}^{n} |\mathbf{a}_i| \tag{34}$$

Also we will use the following sum bounds:

$$\sum_{i=1}^{n} i = \frac{n(n+1)}{2} \tag{35}$$

and

$$\sum_{i=a}^{b} \frac{1}{2\sqrt{i}} \leq \int_{a-1}^{b} \frac{1}{2\sqrt{x}} dx = \sqrt{b} - \sqrt{a-1} \leq \sqrt{b-a+1} \,. \tag{36}$$

We start with summing (13) along iterations:

$$\sum_{t=1}^{T} < \mathbf{x}_t - \mathbf{x}^*, \tilde{g}_t >$$

$$= \sum_{t=1}^{T} \frac{1}{2} \eta_t ||\bar{g}_t||^2 + \sum_{t=1}^{T} \frac{D(\mathbf{x}^*||\mathbf{x}_t) - D(\mathbf{x}^*||\mathbf{x}_{t+1})}{\eta_t} + \sum_{t=1}^{T} < \mathbf{x}_t - \mathbf{x}^*, \tilde{g}_t - \bar{g}_t >$$

$$= \sum_{t=1}^{T} \left[ \frac{1}{2} \eta_t ||\bar{g}_t||^2 \right] + \frac{D(\mathbf{x}^*||\mathbf{x}_1)}{\eta_1} - \frac{D(\mathbf{x}^*||\mathbf{x}_{T+1})}{\eta_T} + \sum_{t=2}^{T} \left[ D(\mathbf{x}^*||\mathbf{x}_t) \left( \frac{1}{\eta_t} - \frac{1}{\eta_{t-1}} \right) \right]$$

$$+ \sum_{t=1}^{T} < \mathbf{x}_t - \mathbf{x}^*, \tilde{g}_t - \bar{g}_t > \,. \tag{37}$$

Next, we borrow the derivation of expressions (38) and (39) from the proof of Theorem 2, Zinkevich et al. (2009a). Note, however, that we added a new term to (37) – the last term that is specific to Algorithm 2. This term does not appear in the proof of Theorem 2, Zinkevich et al. (2009a). By the Lipschitz property of gradients and the definition of $\eta_t$, we can bound the first summand of the above regret expression via

$$\sum_{t=1}^{T} \frac{1}{2} \eta_t ||\bar{g}_t||^2 \leq \sum_{t=1}^{T} \frac{1}{2} \eta_t \frac{1}{|W|} \sum_{w \in W} ||g_{t,w}||^2 \leq \sum_{t=1}^{T} \frac{1}{2} \eta_t L^2 = \sum_{t=1}^{T} \frac{1}{2} \frac{\sigma}{\sqrt{t}} L^2 \leq \sigma L^2 \sqrt{T} \,. \tag{38}$$

Also

$$\frac{D(\mathbf{x}^*||\mathbf{x}_1)}{\eta_1} + \sum_{t=2}^{T} D(\mathbf{x}^*||\mathbf{x}_t) \left[ \frac{1}{\eta_t} - \frac{1}{\eta_{t-1}} \right] \leq \frac{F^2}{\sigma} \sqrt{T} \,. \tag{39}$$

We omit the negative factor $-\frac{D(\mathbf{x}^*||\mathbf{x}_{T+1})}{\eta_T}$. Now we start the analysis of the last term in (37), which is new and specific to Algorithm 2. Using (34), we bound the last summand of (37)

$$\left| \sum_{t=1}^{T} < \mathbf{x}_t - \mathbf{x}^*, \bar{g}_t - \tilde{g}_t > \right| \leq \sum_{t=1}^{T} ||\mathbf{x}_t - \mathbf{x}^*|| \cdot ||\tilde{g}_t - \bar{g}_t|| \,. \tag{40}$$

Substituting (38), (39) and (40) into (37), we get

$$R[X] \leq \sigma L^2 \sqrt{T} + \frac{F^2}{\sigma} \sqrt{T} + \sum_{t=1}^{T} ||\mathbf{x}_t - \mathbf{x}^*|| \cdot ||\tilde{g}_t - \bar{g}_t|| \,, \tag{41}$$

Next, we proceed with bounding the last multiplicative factor in the RHS of (40)

$$||\tilde{g}_t - \overline{g}_t|| = \left|\left| \frac{1}{|W|} \sum_{w \in W} g_{t,w} - \frac{1}{|W|} \sum_{w \in W} \nabla f_{t,w}(\mathbf{x}_t) \right|\right| = \frac{1}{|W|} \left|\left| \sum_{w \in W} (g_{t,w} - \nabla f_{t,w}(\mathbf{x}_t)) \right|\right|$$

$$\leq \frac{1}{|W|} \sum_{w \in W} ||g_{t,w} - \nabla f_{t,w}(\mathbf{x}_t)|| \leq \frac{H}{|W|} \sum_{w \in W} ||\mathbf{x}_{t,w} - \mathbf{x}_t|| . \tag{42}$$

In (42) we used inequality (34) and (14). Also note that if $t$ is the last iteration in a cycle, $g_{t,w}$ is computed on the parameter vector before the reset operation (7) is applied. Now we bound each summand in (42). For $t = i\beta + j$

$$\mathbf{x}_{i\beta+j,w} - \mathbf{x}_{i\beta+j} = \mathbf{x}_{i\beta,w} + \sum_{k=0}^{j-1} -\eta_{i\beta+k} g_{i\beta+k,w} - \frac{1}{|W|} \sum_{w' \in W} \left( \mathbf{x}_{i\beta,w'} + \sum_{k=0}^{j-1} -\eta_{i\beta+k} g_{i\beta+k,w'} \right)$$

$$= \frac{1}{|W|} \sum_{w' \in W} (\mathbf{x}_{i\beta,w} - \mathbf{x}_{i\beta,w'}) - \sum_{k=0}^{j-1} \eta_{i\beta+k} \frac{1}{|W|} \sum_{w' \in W} (g_{i\beta+k,w} - g_{i\beta+k,w'}) \tag{43}$$

Using Lemma 3.3, the assumption that gradients of cost functions are bounded by $L$, and learning rate decreases as iterations increase,

$$||\mathbf{x}_{i\beta+j,w} - \mathbf{x}_{i\beta+j}|| \leq L\tau\eta_{(i-1)\beta} + \sum_{k=0}^{j-1} \eta_{i\beta} \frac{1}{|W|} \sum_{w' \in W} 2L \leq 2L\tau\eta_{(i-1)\beta} . \tag{44}$$

Substituting (44) into (42), gives

$$||\overline{g}_{i\beta+j} - \tilde{g}_{i\beta+j}|| \leq \frac{H}{|W|} \sum_{w \in W} 2L\tau\eta_{(i-1)\beta} \leq 2\tau L H \eta_{(i-1)\beta} . \tag{45}$$

Now, using the fact that learning rate is a decreasing function and the assumption that distance between any pair of points is bounded by $F^2$, we substitute (45) into (40) to get

$$\left| \sum_{t=1}^{T} < \mathbf{x}_t - \mathbf{x}^*, \overline{g}_t - \tilde{g}_t > \right| \leq \sum_{t=1}^{T} 2F\tau L H \eta_{t-2\tau} = 2F\tau L H \sum_{t=1}^{T} \eta_{t-2\tau} . \tag{46}$$

Note that for $t < 1$, we use $\eta_t = 1$. Next, we separate the iteration $t$ into two ranges: $t = 1, \ldots, 2\tau$ and $t > 2\tau$

$$\sum_{t=1}^{2\tau} \eta_{t-2\tau} + \sum_{t=1}^{T-2\tau} \eta_t = 2\tau + 2\sigma \sum_{t=1}^{T-2\tau} \frac{1}{2\sqrt{t}} \leq 2\tau + 2\sigma\sqrt{T - 2\tau} . \tag{47}$$

Substituting (47) into (46)

$$\left| \sum_{t=1}^{T} < \mathbf{x}_t - \mathbf{x}^*, \overline{g}_t - \tilde{g}_t > \right| \leq 2\tau F L H (2\tau + 2\sigma\sqrt{T - 2\tau}) . \tag{48}$$

Substituting (38), (39) and (48) into (37), we prove (15). Using $\sigma^2 = \frac{F^2}{2\tau L^2}$ in (15), we prove (16). □

Proof of Lemma 3.6.

*Proof.* We prove the lemma by induction on $j$. For $j = 0$, the statement of lemma holds. Now assume that the lemma holds for $j - 1$. Then

$$||\mathbf{x}_{i\beta+j,w} - \mathbf{x}_{i\beta+j,w'}|| = ||\mathbf{x}_{i\beta+j-1,w} - \mathbf{x}_{i\beta+j-1,w'}$$
$$- \eta_{i\beta+j-1}(g^*(\mathbf{x}_{i\beta+j-1,w}) - g^*(\mathbf{x}_{i\beta+j-1,w'}))$$
$$- \eta_{i\beta+j-1}(e_{i\beta+j-1,w} - e_{i\beta+j-1,w'})||$$
$$\leq ||\mathbf{x}_{i\beta+j-1,w} - \mathbf{x}_{i\beta+j-1,w'}|| + \eta_{i\beta+j-1}||g^*(\mathbf{x}_{i\beta+j-1,w}) - g^*(\mathbf{x}_{i\beta+j-1,w'})||$$
$$+ \eta_{i\beta+j-1}||e_{i\beta+j-1,w} - e_{i\beta+j-1,w'}|| .$$

We use (14) in the above expression

$$||\mathbf{x}_{i\beta+j,w} - \mathbf{x}_{i\beta+j,w'}||$$
$$\leq (\eta_{i\beta+j-1}H + 1)||\mathbf{x}_{i\beta+j-1,w} - \mathbf{x}_{i\beta+j-1,w'}|| + \eta_{i\beta+j-1}||e_{i\beta+j-1,w} - e_{i\beta+j-1,w'}||$$

Now, applying the inductive assumption

$$||\mathbf{x}_{i\beta+j,w} - \mathbf{x}_{i\beta+j,w'}|| \leq (\eta_{i\beta+j-1}H + 1)||\mathbf{x}_{i\beta,w} - \mathbf{x}_{i\beta,w'}|| \cdot \prod_{k=0}^{j-2}(\eta_{i\beta+k}H + 1)$$

$$+ (\eta_{i\beta+j-1}H + 1)\sum_{k=0}^{j-2}\eta_{i\beta+k}||e_{i\beta+k,w} - e'_{i\beta+k,w'}|| \cdot \prod_{m=k+1}^{j-2}(\eta_{i\beta+m}H + 1)$$

$$+ \eta_{i\beta+j-1}||e_{i\beta+j-1,w} - e_{i\beta+j-1,w'}|| \tag{49}$$

Inserting $(\eta_{i\tau+j-1}H + 1)$ into product in the first summand in (49) amd into the sum in the second summand,

$$||\mathbf{x}_{i\beta+j,w} - \mathbf{x}_{i\beta+j,w'}|| \leq ||\mathbf{x}_{i\beta,w} - \mathbf{x}_{i\beta,w'}||\prod_{k=0}^{j-1}(\eta_{i\beta+k}H + 1)$$

$$+ \sum_{k=0}^{j-2}\eta_{i\beta+k}||e_{i\beta+k,w} - e'_{i\beta+k,w'}|| \cdot \prod_{m=k+1}^{j-1}(\eta_{i\beta+m}H + 1) + \eta_{i\beta+j-1}||e_{i\beta+j-1,w} - e_{i\beta+j-1,w'}||$$

$$\tag{50}$$

Now, note that the last summand in (50) corresponds to the summand in (18) for $k = j - 1$. This completes the proof. $\square$

Proof of Lemma 3.7.

*Proof.* First note that from (19) it follows that

$$\eta_t\beta H \leq 0.1 . \tag{51}$$

Since $\eta_i$ shrinks down, as $i$ grows up, and $\eta_t H + 1 > 1$, (18) yields

$$||\mathbf{x}_{i\beta+j,w} - \mathbf{x}_{i\beta+j,w'}|| \leq (\eta_{i\beta}H + 1)^{\beta}$$

$$\cdot \left(||\mathbf{x}_{i\beta,w} - \mathbf{x}_{i\beta,w'}|| + \eta_{i\beta}\sum_{k=0}^{\beta-1}||e_{i\beta+k,w} - e_{i\beta+k,w'}||\right) . \tag{52}$$

After the expansion of $(\eta_{i\beta}H + 1)^{\beta}$ in (52) into Taylor sequence and applying a simple algebra

$$(\eta_{i\beta}H + 1)^{\beta} \leq 1 + \beta\eta_{i\beta}H + (\beta\eta_{i\beta}H)^2(1 - \beta\eta_{i\beta}H)^{-1} \tag{53}$$

From (51) we can bound

$$(\eta_{i\beta}H + 1)^{\beta} \leq 1.2 . \tag{54}$$

Assigning (54) into (52),

$$||\mathbf{x}_{i\beta+j,w} - \mathbf{x}_{i\beta+j,w'}|| \leq 1.2 \cdot ||\mathbf{x}_{i\beta,w} - \mathbf{x}_{i\beta,w'}|| + 1.2 \cdot \eta_{i\beta}\sum_{k=0}^{\beta-1}||e_{i\beta+k,w} - e_{i\beta+k,w'}|| . \tag{55}$$

Since $e_{i\tau+k,w}$ and $e_{i\tau+k,w'}$ are i.i.d. with mean 0,

$$E||e_{i\beta+k,w} - e_{i\beta+k,w'}|| = var(e_{i\beta+k,w} - e_{i\beta+k,w'}) = 2var(e_{i\beta+k,w}) . \tag{56}$$

Taking expectation of the both sides of (55), substituting (56) into the resulting inequality and using the assumption on variance of random variables $e_{t,w}$, we complete the proof. $\square$

Proof of Lemma 3.8.

*Proof.* From Lemma 3.3,

$$||\mathbf{x}_{i_0\beta,w} - \mathbf{x}_{i_0\beta,w'}|| \leq 2L\tau\eta_{(i_0-1)\beta} . \tag{57}$$

From (22), each cycle $j$ after iteration $i_0\beta$, reduces the distance between $w$ and $w'$ by factor of 0.2, while adding to the distance the value of $1.2 \cdot \eta_{i\beta+j}\tau\bar{s}$. This means that after the number of cycles $j_0$

$$j_0 = \log \frac{1.2 \cdot \eta_{i_0\beta}\tau\bar{s}}{2L\tau\eta_{(i_0-1)\beta}} = \log \frac{\bar{s}}{L} ,$$

for any $i \geq i_0 + j_0$, the first summand in (22) gets bounded by $0.8 \cdot \eta_{i\beta}\tau\bar{s}$. $\qquad\square$

Proof of Theorem 3.9.

*Proof.* To prove this theorem, we follow the proof of Theorem 3.5 and develop an alternative bound on $||\mathbf{x}_{t,w} - \mathbf{x}_t||$ in (42).

We will split the sum in (40) into two ranges of $t$: $t < t_0 + j_0\beta$ and $t \geq t_0 + j_0\beta$ for $t_0, j_0$, defined in (19) and (23) respectively. To simplify notation, from (23), we can assume that $j_0$ is a constant and we can assume that $t_0 + j_0 \leq 2t_0$. For $t < 2t_0$, we use (48) to show

$$\sum_{t=1}^{2t_0-1} ||\mathbf{x}_t - \mathbf{x}^*|| \cdot ||\bar{g}_t - \tilde{g}_t|| \leq 2\tau FLH(2\tau + 2\sigma\sqrt{2t_0 - 2\tau}) . \tag{58}$$

For $t \geq 2t_0$, we first observe

$$||x_{t,w} - x_t|| = \frac{1}{|W|} \left|\left| \sum_{w' \in W} (x_{t,w} - x_{t,w'}) \right|\right| \leq \frac{1}{|W|} \sum_{w' \in W} ||x_{t,w} - x_{t,w'}|| . \tag{59}$$

Next we use (24) to bound the above expression.

$$E||x_{t,w} - x_t|| \leq 2\eta_{t-\tau}\tau\bar{s} . \tag{60}$$

Substituting this bound into (42), we get

$$E||\bar{g}_t - \tilde{g}_t|| \leq 2H\eta_{t-\tau}\tau\bar{s} . \tag{61}$$

Using (61) in (40) for $t \geq 2t_0$

$$E\left| \sum_{t=2t_0}^{T} < \mathbf{x}_t - \mathbf{x}^*, \bar{g}_t - \tilde{g}_t > \right| \leq \sum_{t=2t_0}^{T} E(||\mathbf{x}_t - \mathbf{x}^*|| \cdot ||\bar{g}_t - \tilde{g}_t||) \leq \sum_{t=2t_0}^{T} FE||\bar{g}_t - \tilde{g}_t|| >$$

$$\leq F \sum_{t=t_0}^{T} 2H\eta_{t-\tau}\tau\bar{s} \leq 4FH\sigma\sqrt{T}\tau\bar{s} . \tag{62}$$

Combining (58) and (62) in (40) and using the definition (19) of $t_0$

$$E\left| \sum_{t=1}^{T} < \mathbf{x}_t - \mathbf{x}^*, \bar{g}_t - \tilde{g}_t > \right| \leq 4\tau^2 FLH(1 + 2\sigma^2 H) + 4FH\sigma\sqrt{T}\tau\bar{s} \tag{63}$$

Using the assumption (25), and substituting into (41) we prove (26). Finally we use value $\sigma = \frac{F}{L}$ to prove (27). $\qquad\square$

