# OpenReview forum: "Asynchronous SGD without gradient delay for efficient distributed training"
_ICLR.cc/2019/Conference_

### Official Review · AnonReviewer3 · 2018-10-17
**I don't understand why the proposed method is an asynchronous method**

**Rating:** 4
**Confidence:** 5

**Review:**

This paper tries to propose a so-called hybrid algorithm to eliminate the gradient delay of asynchronous methods. The authors propose algorithm 1 and a simplified version algorithm 2 and prove the convergence of algorithm 2 in the paper.  The paper is very hard to follow, especially the algorithm description part. What I can understand is that the authors want to let the fast workers do more local updates until the computation in the slowest worker is done. The idea is similar to EASGD except that it forces the workers to communicate the server once the slowest one has completed their job.

The following are my concerns:
1. Do you consider the overhead in constructing the communication between machines? in your method,  workers are keeping notifying servers that they are done with the computation.
2. In Algorithm 1 line 9 and line 23, there are two assignments: x_init =x and x_init=ps.x, is there any conflict?
3. In Algorithm 2,  at line 6 workers wait to receive ps.x, at line 20 server wait for updates. I think there is a bug, and nothing can be received at both ends.
4. The experiments are too weak. There is no comparison between other related methods, such as downpour, easgd.
5. The authors test resnet50 on cifar10,  however, there is no accuracy result. They show the result by using googlenet, why not resnet50? I am curious about the experimental settings.

Above all, the paper is hard to follow and the idea is very trivial. Experiments in the paper are also very weak.

---

### Official Review · AnonReviewer1 · 2018-10-30
**missing references, theory is not novel, experiments are not sufficient**

**Rating:** 4
**Confidence:** 4

**Review:**

The paper proposes an algorithm to restrict the staleness in ASGD (asynchronous SGD), and also provides theoretical analysis. This is an interesting and important topic. However, I do not feel that this paper solves the fundamental issue - the staleness will be still very larger or some workers need to stay idle for a long time in the proposed algorithm if there exists some extremely slow worker. To me, the proposed algorithm is more or less just one implementation of ASGD, rather than a new algorithm. The key trick in the algorithm is collecting all workers' gradients in the master machine and update them at once, while hard limiting the number of updates in each worker. The theoretical analysis is not brand new. The
line 6 in Algorithm 1 makes the delay a random variable related to the speed of a worker. The faster a worker is, the larger the tau is, which invalidates the assumption implicitly used in the theoretical analysis.

The experiment is done with up to 4 workers, which is not sufficient to validate the advantages of the proposed algorithm compared to state of the art ASGD algorithms. The comparison to other ASGD implementations is also missing, such as Hogwild! and Allreduce.

In addition, I am so surprised that this paper only have 10 references (the last one is duplicated). The literature review is quite shallow and many important work about ASGD are missing, e.g.,

- Parallel and distributed computation: numerical methods, 1989.
- Distributed delayed stochastic optimization, NIPS 2011.
- Hogwild!, NIPS 2011
- Asynchronous Parallel Stochastic Gradient for Nonconvex Optimization, NIPS 2015
- An asynchronous mini-batch algorithm for regularized stochastic optimization, 2016.

---

### Official Review · AnonReviewer2 · 2018-11-03
**Interesting paper but the contribution seems not be good enough**

**Rating:** 5
**Confidence:** 4

**Review:**

Overall, this paper is well written and clearly present their main contribution.
However, the novel asynchronous distributed algorithm seems not be significant enough.
The delayed gradient condition has been widely discussed, but there are not enough comparison between these variants.

---

### Meta-Review · Area_Chair1 · 2018-12-10
**Not significant contribution and not sufficient experiments**

**Confidence:** 4
**Recommendation:** Reject

**Metareview:**

Improving the staleness of asynchronous SGD is an important topic. This paper proposed an algorithm to restrict the staleness and provided theoretical analysis. However, the reviewers did not consider the proposed algorithm a significant contribution. The paper still did not solve the staleness problem, and it was lack of discussion or experimental comparison with the state of the art ASGD algorithms. Reviewer 3 also found the explanation of the algorithm hard to follow.